# Differences in Park Walking, Comparing the Physically Inactive and Active Groups: Data from mHealth Monitoring System in Seoul

**DOI:** 10.3390/ijerph19010395

**Published:** 2021-12-30

**Authors:** Youngjun Park, Sunjae Lee, Sohyun Park

**Affiliations:** Department of Architecture & Architectural Engineering, Seoul National University, Seoul 08826, Korea; youngjourpark@gmail.com (Y.P.); leesunjae@snu.ac.kr (S.L.)

**Keywords:** urban park, physical inactivity, walking measures, health disparity, mobile health

## Abstract

Despite the overall increase in physical activities and park uses, the discrepancies between physically inactive and active people have increasing widened in recent times. This paper aims to empirically measure the differences in walking activity in urban parks between the physically inactive and active. As for the dataset, 22,744 peoples’ 550,234 walking bouts were collected from the mHealth system of the Seoul government, using the smartphone healthcare app, WalkOn, from September to November 2019, in Seocho-gu district, Seoul, Korea. We classified the physically inactive and active sample groups, based on their regular walking (≥150 min of moderate-to-vigorous walking activity a week), and analyzed their park walking activities. We found that while there was no significant difference in walking measures of non-park walking between the sample groups, the difference did exist in park walking. The park walking average in the physically active group had more steps (*p* = 0.021), longer time (*p* = 0.008), and higher intensity (*p* < 0.001) of walking than that in the inactive group. Each park also revealed differences in its on-site park walking quantity and quality, based on which we could draw the list of ‘well-walked parks’, which held more bouts and more moderate-to-vigorous physical activities (MVPAs) than other parks in Seocho-gu district. This paper addresses how park walking of physically inactive and active people is associated with multiple differences in everyday urban walking.

## 1. Introduction

Given a wide consensus that urban facilities play a role in reducing sedentary behaviors and physical inactivity, urban parks have been a focus as one of the most health-promoting urban facilities [1,2]. The positive associations of walking activity with built environmental features of parks have been recognized [3,4]. Activity-promoting programs in parks, such as park prescriptions, also develops people’s regular physical activities [5]. Research has been conducted on the urban environmental and behavioral aspects of parks in terms of the physical activity, for example, walking activity in park.

Despite the overall increase in physical activity and park use, discrepancies between physically active and inactive people increase, and those who are not physically active have yet to participate in park activities [6,7]. The group of physical inactivity, categorized as not achieving 150 min of moderate-to-vigorous-intensity physical activity per week, needs to be spotlighted in order to make significant health intervention changes [8,9]. Studies have found that demographic factors and park characteristics are associated with park use [3,10], but little is known about physically inactive people who are in need of the health intervention [11]. 

Innovations in physical activity measurement technology [12] and mobile health [13] are supported in the research. In particular, walking assessment has been developed from self-reported methods, such as travel survey, and now objective measurements capable of more quantitative analysis. Global positioning system (GPS), a widely known technology for objective measurement of locations and paths, has contributed to the analysis of pedestrian behaviors. Combined with geographic information system (GIS), many studies have found the characteristics of walking activity in specific urban area: walking in retail districts [14] and central business districts [15]; hiking trails in recreational areas [16]; and park use in urban forest parks [17]. Today, most smartphones are equipped with GPS sensors, and their operating systems detect user’s body movement and accumulate physical activity data. The smartphone-based approach enables easier physical activity assessment from a larger population than other methods [18,19].

This paper aims to measure the differences in walking activities at the urban parks between those individuals who are physically inactive and active, and to identify a list of parks that are likely to encourage people’s moderate-to-vigorous physical activity. For this goal, three objectives are carried out: (1) to classify individuals who are in physically inactive and active groups; (2) to find a significant difference between their walking measures; (3) to identify the well-walked parks based on the analysis of park walking. These findings help re-evaluate the supportive effect of urban parks on different walking activities in the physically inactive and active groups, and thus applicable for targeted interventions by identifying the specific group of physical inactivity. This paper also provides a practical example of applying a smartphone-based monitoring system to health policy, in collaboration with urban environment researchers, computer engineers, and public health officials.

## 2. Research Data and Methods

This study uses physical activity data from the mHealth monitoring system of the Seoul government, assessed by a smartphone healthcare application, WalkOn (Swallaby Inc., Seoul, South Korea). The reason why this app is suitable for research purposes is that, first, it has been officially used by public health centers in Korea; second, it is free of charge, serviced in Korean, and has more than 500,000 active users (about 1% of national population); and third, it is highly applicable to various smartphone devices because it is based on the physical activity assessment algorithm of Android and iOS operating systems. However, its data might be biased because the user percentage is higher in those individuals over the age of 40 and lower in the under 10 age group. Due to the restriction of personal information, we cannot present the prevalence of ages of our dataset. The period of the dataset, from September to November 2019, was the autumn season with abundant outdoor activities and the season before the outbreak of COVID-19, that might have brought an impact on people’s outdoor activities. The research site, Seocho-gu district, is one of 25 administrative districts of Seoul with low obesity prevalence, great walkability, and the largest number of pedestrians [20]. The local government of Seocho-gu district has tried to provide walkable built environments, including its 130 urban parks, and run several walk-promoting campaigns through the mobile app. The dataset of this study was composed of 22,744 app users who recorded 550,234 bouts of walking activity in Seocho-gu for 13 weeks, from September to November 2019. Technical reliability of this data basically follows the confidence level of Android and iOS algorithms that detect physical activity from each device. This is because the raw data of this app has been called from smartphone operating systems.

Research data of this study are mainly composed of three types of information: smartphone-based physical activity data for walking activities; smartphone-based regular walking data for physical inactivity level of subjects; and Seoul urban GIS data for park facilities (https://data.seoul.go.kr/dataList/OA-21129/S/1/datasetView.do (accessed on 27 April 2021)). Our dataset of walking activities, which was an existing dataset from the Seoul mHealth monitoring system, has information of pseudonymized user ID, start time, end time, the number of steps, and GPS coordinates of its path (Appendix A, Table A1). A de-identification process must be applied on research data so that individuals cannot be specified, according to the Personal Information Protection Act in Korea. In order to determine the user’s physical inactivity, the number of weeks that the user had achieved regular walking was measured over a 13-week research period. Park facilities in the GIS dataset, shown in Figure 1 were used to distinguish park walking, as a walking activity in which more than one of GPS coordinates exists within the park area. Our definition of park walking includes a wide range of park-related walking such as walking inside or passing through a park. To identify park walking and non-park walking, we used Python library, GeoPandas version 0.8.1 (https://geopandas.org/en/stable/ (accessed on 19 December 2021)), and Matplotlib version 3.5 (https://matplotlib.org/ (accessed on 19 December 2021)), to check point-in-polygon whether every GPS points were located in 130 parks in Seocho-gu or not (Appendix A, Table A2). As a result, 56,651 bouts of park walking (10.26%) and 493,583 bouts of non-park walking (89.74%) were identified from 550,234 walking bouts.

For understanding, the terms used in this study are summarized as follows:

Physical inactivity: Public health physical activity guidelines [2] recommend that people engage in sufficient levels of moderate-to-vigorous physical activity (MVPA) for older adults, at least 150 min/week, and include “brisk walking” as a primary example of an appropriate activity [21,22,23].Regular walking: at least 150 min/week of walking activity. This criteria is officially recommended by public health professionals to make it easier for the public to understand and promote their physical activities [24,25].Physically inactive/active sample group: Based on the regular walking of each person, this study categorized people into physically inactive/active sample group.Park walking and non-park walking: a walking activity which does/does not pass through park areas in research site. To identify park walking activities, we coded a point-in-polygon algorithm in Python to determine whether every GPS points of walking activity were located in park or not.Walking measures: quantitative information of walking activity, including accelerometer counts, GPS coordinates, and timestamp, were recorded by smartphone device. The application of the mHealth system can calculate those numbers into several walking measures, such as steps/bout, duration/bout, and cadence/bout.

In order to find the differences in walking measures between physically inactive and active people, we designed this research into three processes: (1) selecting sample groups; (2) comparing sample groups; and (3) finding certain parks based on the differences between sample groups. First, sample selection classified the physical inactivity level of each user by the number of weeks of regular walking (>150 min/week of moderate-to-vigorous walking activities). Two distinct sample groups, who did/did not achieve regular walking every week in the research period, were selected.

Second, we compared the walking measures of the sample groups’ park walking and non-park walking. Walking measures of each group, for example, the number of bouts, steps, duration, and cadence, were calculated from the dataset. In the result of this study, the average walking measures of each group were compared, and a significant statistical difference between the two groups was tested by using Mann–Whitney U test.

Third, based on the differences in overall walking activities between each sample groups, it is necessary to figure out the apportioned differences at each park. Considering the health intervention for physically inactive people, we analyzed whether a certain park had better walking measures. The list of ‘well-walked parks’ was drawn based on our spatial analysis. Considering the criteria of physical inactivity, 150 min/week of MVPA, both volume and intensity of park walking are essential for physically inactive people [2,6]. We found mostly-walked parks to have a greater number of walking bouts. Mostly-walked parks in each sample group were compared to find differences in the quantity of park walking. In addition to bouts of park walking, we also figured out several parks that have higher level of walking intensity. Detailed procedures and findings in Figure 2 are explained in the following Results section.

## 3. Results

### 3.1. Sample Selection: Classifying the Physically Inactive and Active Groups

Sample selection, for the first step, needed to find physically inactive and active people based on their regular walking (≥150 min/week of moderate-to-vigorous walking). The mHealth monitoring system of Seoul, our data source, evaluates the regular walking of app users from their smartphones every week. Its algorithm (small vibrations, short movements, and light intensity of walking activities are excluded from the regular walking measurement) considers walking activities meeting the criteria of three features (≥1000 steps, ≥10 min, ≥100 steps/min) as MVPA walking [12,26,27]. If an app user had 150 or more minutes of MVPA walking in a week, he or she would be counted as meeting the regular walking in that week [28]. Based on the user’s accomplishment of regular walking in 13 weeks of our research period, we can evaluate each user for how many weeks he or she met the public health recommendation. As shown in Table 1, 339 people who never met the regular walking for 13 weeks were selected as a physically inactive sample group (Group 0, from here on), and 5078 people who successfully met regular walking for all 13 weeks were selected as a sample of the physically active sample group (Group 13). In order to increase the reliability of the sample, we selected the two distinctive sample groups, which had a maximum and minimum of regular walking week, and fully monitored for the entire research period (a portion of data was excluded from the sample selection; 6372 cases that were uncertain because they partially succeeded or failed to meet regular walking, and 10,955 cases that were monitored less than 13 weeks were excluded).

### 3.2. Differences in Walking Activities: Comparing the Selected Sample Groups

In order to compare the walking activities, it is necessary to comprehensively look at three indicators representing different aspects of walking, step, duration, and cadence [29]. Step is a familiar indicator to explain the volume of a walking bout. Duration is another quantitative indicator representing the temporal aspect of a walking bout. Cadence, the number of steps per minute, represents the intensity of walking activity. Considering that the public health recommendation targets physically inactive people to increase moderate-to-vigorous-intensity physical activity for 150 min or more per week, we considered not only the quantity of walking but also the intensity aspect of walking as an important factor in the analysis.

As shown in Table 2, the walking measures of the sample groups’ non-park walking and park walking were carefully calculated, based on which park walking had different walking measures in each group. Considering the average walking measures, park walking of Group 13 showed an average of 2108.13 steps, 26.09 min, and 76.19 steps/min in a single walking bout. Park walking of Group 0 had an average of 1665.96 steps, 24.58 min, and 65.64 steps/min in a single walking bout. In the two groups, the walking measures of park walking seemed higher than those of non-park walking. However, it was not enough to prove the differences in walking measures between the two sample groups.

To statistically confirm a significant difference between the two sample groups, the data should be normally distributed. If not, a non-parametric test is needed. As shown in Figure 3, it looks far from the normal distribution, but close to Pareto distribution (power-law distribution) with a long tale. In order to test the homogeneity of two independent samples’ walking measures, Mann–Whitney U test (Wilcoxon rank-sum test) was conducted as a non-parametric test. The result in Table 3 showed that three walking measures (steps, duration, and cadence) of park walking were significantly different between Group 0 and Group 13, but those measures of non-park walking have no significant difference. Steps of park walking were significantly lower in Group 0 (median = 1169, *n* = 587) compared to Group 13 (median = 1191.5, *n* = 12,260), U = 3,395,179, z = −2.31, *p* = 0.02066. Cadence of park walking was significantly lower in Group 0 (median = 68.07) compared to Group 13 (median = 77.15), U = 2,814,828, z = −8.93, *p* < 0.001. Duration of park walking was significantly higher in Group 0 (median = 18.9) compared to Group 13 (median = 16.7), U = 3,828,546, z = 2.62, *p* = 0.008719. Otherwise, non-park walking in both groups had no significant differences in steps (*p* = 0.3087), duration (*p* = 0.1171), and cadence (*p* = 0.08002). Therefore, non-park walking of the two groups does not show a significant difference in all three indicators of step, duration, and cadence, whereas park walking of the two groups shows a significant difference in all three indicators. Cadence of park walking in the two groups, which is associated with MVPA, shows an extremely significant difference at a confidence level of less than 0.1%.

Highlighted on the cadence, which shows a significant difference in the two groups, the park walking of the two sample groups was divided into light, moderate, and vigorous levels of intensity. The intensity level of walking activity is another factor of the difference between the two groups park’ walking. Using the thresholds of cadence values, we classified the walking activities into the light (<100), moderate (100 ≤ cadence < 130), and vigorous-intensity (≥130) of walking. As shown in Table 4, a total of 587 park walking in Group 0 showed the distribution of light (91.82%), moderate (7.84%), and vigorous activity (0.34%). On the other hand, in Group 13, a total of 12,260 park walking, light (80.80%), moderate (18.03%), and vigorous activity (1.17%) showed a higher bout ratio of moderate-vigorous walking activity (MVPA). Steps and duration of Group 13 also had greater measures in every intensity level than those of Group 0. It can be inferred that Group 13 walked at the parks with better quality of walking activities than Group 0 did.

After all, between Group 13 and Group 0, there was no significant difference between non-park walking of Group 0 and Group 13, but there was a significant difference in park walking. Taking a more in-depth look at the cadence of park walking, Group 13 walked with a higher level of intensity than Group 0, when they walked in the park. This could be evidence that park walking activities will be helpful in increasing the walking measures of people, and that physically active individuals may have better walking measures than inactive individuals.

### 3.3. Differences in Parks: Finding ‘Well-Walked Park’

Based on the differences in park walking of the two groups, we now examine the differences in spatial aspects. Among the 130 parks in Seocho-gu district, we tried to find ‘well-walked parks’, which provide sufficient MVPAs of both Group 13 and Group 0. A plain indicator for finding parks that hold health-related physical activities might be the number of walking trips, but a more critical indicator is needed to examine the intensity of walking activities. While ‘mostly-walked park’ represents the quantity of park walking, ‘intensely-walked park’ represents the quality of park walking. In order to find ‘well-walked parks’, we firstly filtered the ‘mostly-walked park’ and figured out ‘intensely-walked parks’ with a higher ratio of MVPAs among them.

We listed in a ranking order by the number of bouts in each sample group. Table 5 shows a list of the top 20 of ‘mostly-walked parks’ with the walking measures of three sample groups, Group 0, Group 13, and Group All. Comparing the top 20 of ‘mostly-walked parks’ of sample groups, we can categorize those parks into three common types, as shown in Table 6. The three types of ‘mostly-walked parks’, with their parkID (parkIDs were from the serial number of Seoul Urban GIS dataset. The first-21-digit of raw data, ‘11650UQ153PS201407214′, which is entitled as same as every park in Seocho-gu, was converted to ‘park’ + ‘000′ in the table)s and codes (we named a new code (e.g., 3-A), representing its type of MWP, next to the parkIDs to avoid confusion), consists of Type MWP-3 (common in the list of Group all, Group 13, and Group 0), Type MWP-2 (common in Group all, Group 13) and Type MWP-1 (appeared only in Group 0). MWP-3 had 11 parks, hosting a great number of walking bouts in every sample group. Group 0 not only recorded a large number of bouts in these parks, but also showed great measures in terms of step, duration, and cadence. MWP-2 consisted of five parks in which Group 0 had not as many walking bouts, unlike other groups. This might be related to the walking differences, in that Group 0 tended not to walk, but Group 13 and Group All seem willing to walk. MWP-1, which was only listed in Group 0, might represents extraordinary bouts of Group 0′s park walking. This result, however, must be interpreted with caution due to the small sample size of Group 0.

Furthermore, we analyzed the intensity of walking activity to examine which parks had more moderate-to-vigorous-intensity level of park walking (MVPA). We already estimated the percentage of park walking that was MVPA in each sample group, described in Table 2. The average bouts ratio of MVPA in each group’s park walking was 8.18% in Group 0, 19.20% in Group 13, and 14.95% in Group All. Depending on the average of MVPA%, we evaluated which intensity level of park walking was prevalent in each park. Higher MVPA% at a certain park than the average of park walking meant that people were likely to walk ‘more intensely’ at that park. Table 7 shows whether the prevalent intensity of each park is light or moderate-vigorous in the sample groups. As shown in Table 8, it was possible to categorize seven parks (3-F, 3-G, 3-J, 2-E, 1-A, 1-D, and 1-F) as ‘intensely-walked parks’ which had a higher percentage of MVPA park walking in both Group All and Group 13 (Group 0 were excluded in this comparison, because bouts of each park were too little (<10) in Group 0). On the other hand, 13 parks in which LPA was more prevalent could be sorted as less-intensely-walked parks. Only three parks (3-D, 2-D, and 1-C) have a different prevalent intensity in Group All and Group 13. Figure 4 shows the locations of the three types of ‘mostly-walked parks’ and emphasizes ‘intensely-walked parks’ in Seocho-gu district. 

## 4. Conclusions

The current widespread use of smartphones along with the evolving mHealth monitoring technology enable more abundant data collection as well as detailed analysis for environmental health. In Korea, which has a high level of smartphone ownership (first ranked country, with 95% of smartphone ownership [30]. https://www.pewresearch.org/global/2019/02/05/smartphone-ownership-is-growing-rapidly-around-the-world-but-not-always-equally/ (accessed on 15 August 2021)), smartphone-based research has been increased to explore the walking activities of diverse populations, including vulnerable groups in various urban spaces. A large number of datasets, walking activities from the mHealth monitoring system of Seoul, and park data from the urban planning information system of Seoul, allow us to explore the differences in park walking between the physically inactive and active groups. This study, as an analysis from existing data, has several limitations: uncontrollable conditions of data collection, de-identified information about subjects, and probability of self-selection. Nevertheless, smartphone-based research has promising potential to explore with careful and proper protocols. Researchers look forward to further studies on the global protocol and methodology for smartphone-assessed physical activities.

This paper contributes to a better understanding of the walking activities of physically active and inactive groups in specific urban parks. Empirical contrasts between the physically inactive and active, park walking and non-park walking, mostly-walked parks and intensely-walked parks, are keywords for interpreting the differences of our findings. While previous approaches to the vulnerable group were identified in terms of demographic or socioeconomic variables, this study identified physically inactive people without any demographic information. The findings in this study demonstrated that the physically inactive and active groups had disparity in their park walking. Significant differences between both groups resulted from their park walking, for example, the physically active group had better walking indicators, including more steps, longer time, and higher intensity in park walking, than the inactive group. This is evidence that parks can bring more health benefits to people who do not meet the public health recommendation of regular walking. This study then drew the newly defined ‘well-walked park’, in which most people walk and do so intensely. The finding of the ‘well-walked park’ is a solid foundation for promoting urban places that contain not only quantitative but also qualitative aspects of walking activities. In order to meet 150 min of MVPA as required by the public health recommendation, it is essential to increase the volume of walking activities as well as to increase the intensity of walking activities. As a result, we established the substantial list of ‘well-walked parks’, which brought more bouts and higher intensity. 

This study reflects collaborative works that have been carried out among a local government’s public health office, a university’s urban design and planning lab, and a private sector’s start-up company for the mobile healthcare system. For a common goal of promoting citizen’s physical activities in the urban environment, the three bodies work together to develop and publicize smartphone healthcare apps, collect and monitor data, and conduct research for policy implications. Starting with the results of this study, this type of collaboration is necessary to explore more evidence for activity-supportive built environments based on walking activities empirically measured in urban spaces.

Physical inactivity is a global public health issue, and various policies are being discussed so that people can have more MVPA and reduce their sedentary behavior. This study implies that park walking could provide better walking measures with more steps, duration, and cadence, which are recommended for the physically inactive group. Parks are necessary as urban infrastructure to bring both quantity and quality of walking activities. Despite the overall increase of park areas, the health disparity on physically inactive people gets worse [31]. Empirical monitoring for park spaces acting as public goods means the built environment and health professionals now strategically cooperate with targeted interventions to make changes for the physically inactive. Differences in park walking could suggest evidence to distinguish the advantage of parks, with respect to which they are easy for park walking and favorable for walking longer, and which are effective to increase MVPAs. It could suggest the implication for practice to re-design a park with fewer walking bouts or to run a walk-promoting program at a park with high MVPAs. We hope our collaborative understanding of differences in park walking contributes to providing more supportive built environments for physically inactive people.

## Figures and Tables

**Figure 1 ijerph-19-00395-f001:**
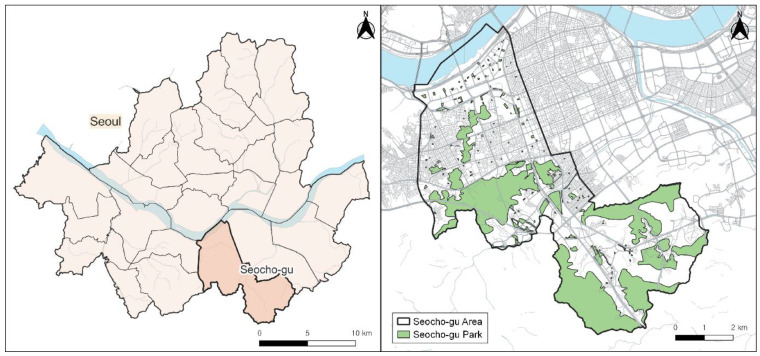
Area map of Seocho-gu district and its parks.

**Figure 2 ijerph-19-00395-f002:**
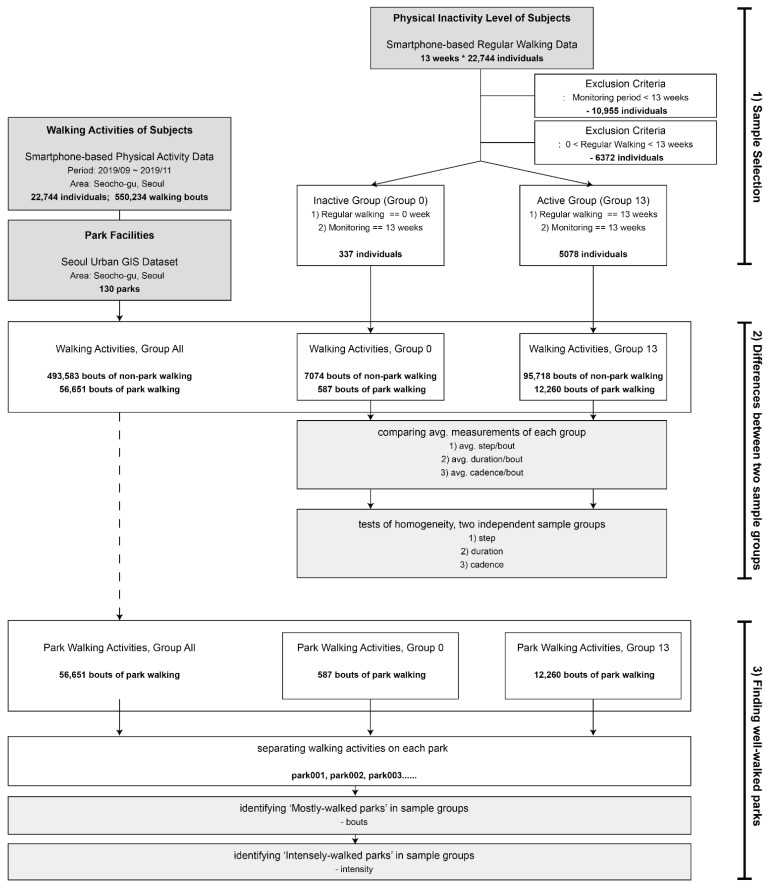
Research Design.

**Figure 3 ijerph-19-00395-f003:**
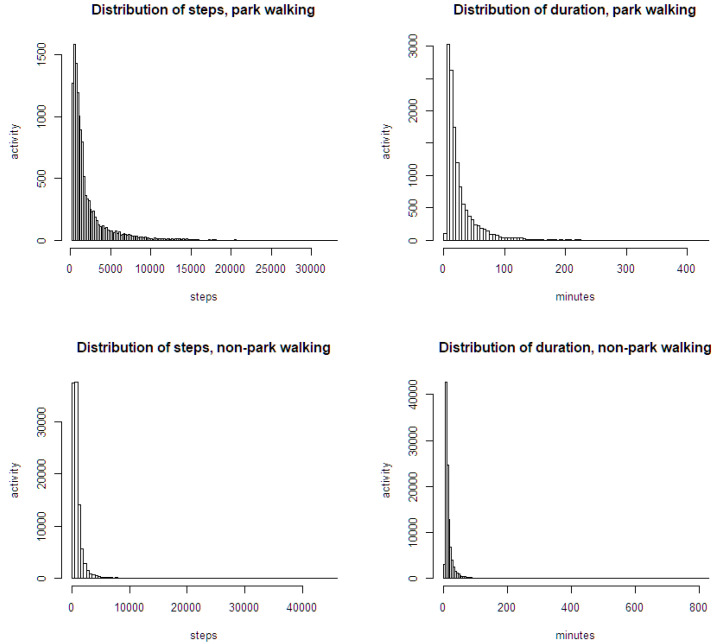
Distribution of steps and duration of walking bouts.

**Figure 4 ijerph-19-00395-f004:**
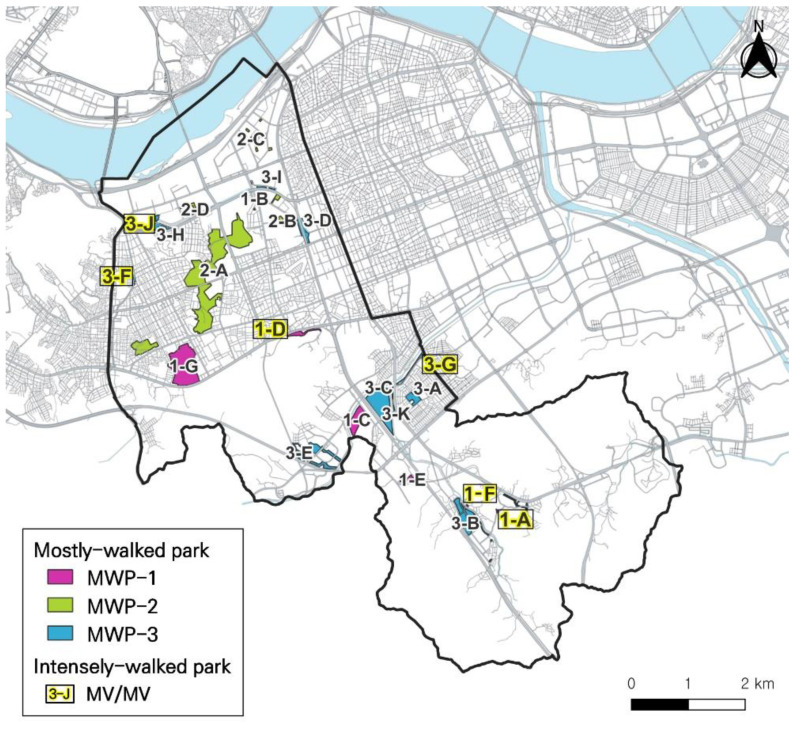
Map visualization of ‘well-walked parks’.

**Table 1 ijerph-19-00395-t001:** Distribution of regular walking by week and sample selection.

Category	Monitoring (Week)	Regular Walking (Week)	People	Sample Selection
Total	-	-	*n* = 22,744	Total Population (Group All)
Monitored for entire research period	13	0	339	Sample of the physically inactive (Group 0)
	13	1	314	-
	13	2	263	-
	13	3	270	-
	13	4	309	-
	13	5	299	-
	13	6	316	-
	13	7	353	-
	13	8	415	-
	13	9	495	-
	13	10	667	-
	13	11	907	-
	13	12	1764	-
	13	13	5078	Sample of the physically active (Group 13)
Monitored less than 13 weeks	1~12	0~12	10,955	-

**Table 2 ijerph-19-00395-t002:** Comparing average measures of sample groups’ park walking and non-park walking.

Sample	Activity	People	Bouts	Bouts/Person (Bout)	Step/Bout (Step)	Duration/Bout (Min)	Cadence/Bout (Step/Min)
Group 0	non-park walking	339	7074	20.99	960.38	15.08	63.68
	park walking	59	587	9.95	1665.96	24.58	65.64
Group 13	non-park walking	5078	95,718	19.02	984.01	15.25	65.54
	park walking	1078	12,260	11.37	2108.13	26.09	76.19
Group All	non-park walking	22,744	493,583	21.7	946.84	15.36	61.64
	park walking	5338	56,651	10.61	1820.07	24.29	72.87

**Table 3 ijerph-19-00395-t003:** Mann–Whitney u tests for two independent sample groups.

Type	Measures	Median of Group 0	Median of Group 13	U	Z	*p*-Value
park walking	step	1169 (*n* = 587)	1191.5 (*n* = 12,260)	3,395,179	−2.31	0.02066 *
	duration	18.9 (*n* = 587)	16.67 (*n* = 12,260)	3,828,546	2.62	0.008719 **
	cadence	68.07 (*n* = 587)	77.15 (*n* = 12,260)	2,814,828	−8.93	<0.001 ***
non-park walking	step	643 (*n* = 7074)	647 (*n* = 95,718)	341,006,274	1.02	0.3087
	duration	11.06 (*n* = 7074)	10.95 (*n* = 95,718)	342,328,097	1.57	0.1171
	cadence	61.83 (*n* = 7074)	62.75 (*n* = 95,718)	334,338,611	−1.75	0.08002

*** *p* < 0.001; ** *p* < 0.01; * *p* < 0.05.

**Table 4 ijerph-19-00395-t004:** Comparing measures of sample groups’ park walking, categorized by the level of intensity.

Sample	Intensity	People (*n*)	Bouts (*n*)	Bouts Ratio (%)	Step (Step)	Duration (Min)	Cadence (Step/Min)
Group 0	Light	59	539	91.82	1490.69	24.14	61.75
	Moderate	14	46	7.84	3271.63	30.51	107.22
	Vigorous	2	2	0.34	1284.51	8.15	157.61
Group 13	Light	1026	9906	80.8	1590.7	23.54	67.56
	Moderate	277	2211	18.03	4119.27	37.24	110.63
	Vigorous	51	143	1.17	4255.64	29.99	141.92
Group All	Light	5109	48,180	85.05	1566.56	23.40	65.99
	Moderate	1364	7949	14.03	3258.72	29.78	109.50
	Vigorous	205	522	0.92	3312.20	22.81	149.50

**Table 5 ijerph-19-00395-t005:** Comparing TOP 20 ‘mostly-walked parks’.

Group All					Group 0					Group 13				
Park ID	Bouts (*n*)	Steps (*n*)	Duration (Min)	Cadence (Steps/Min)	Park ID	Bouts (*n*)	Steps (*n*)	Duration (Min)	Cadence (Steps/Min)	Park ID	Bouts (*n*)	Steps (*n*)	Duration (Min)	Cadence (Steps/Min)
park260 ***	5058	1492.36	23.79	69.26	park286 ***	116	1306.59	19.23	66.77	park260 ***	1188	1644.06	22.25	73.59
park330	4400	1786.98	23.73	72	park201	62	769.02	19.43	40.61	park168 ***	841	1266.69	17.55	73.35
park300 **	3337	1122.06	16.25	68.99	park272	55	1518.11	20.4	79.3	park286 ***	822	1553.8	20.98	74
park286 ***	2933	1747.92	23.79	72.2	park312 ***	33	2135.52	27.29	76.6	park330 **	718	2283.28	28.24	75.28
park168 ***	2821	1162.03	18.52	66.56	park248 *	22	759.77	13.59	51.47	park300 **	692	860.48	13.07	63.87
park259 ***	2167	2300.99	29.95	75.37	park122 ***	19	2526.74	30.74	80.83	park204 **	477	1027.21	14.07	76.85
park204 **	2126	1223.54	17.81	72.52	park168 ***	17	834.71	16.94	54.69	park312 ***	462	1414.05	18.22	72.03
park312 ***	2022	1499.36	20.11	71.01	park118 *	16	1930.75	30.61	69.52	park122 ***	400	2122.15	25.29	82.92
park122 ***	1727	1877.45	24.73	73.04	park219 ***	16	1483.88	20.53	67.8	park244 **	386	5363.74	50.84	94.68
park264 ***	1645	4483.46	47.54	88.12	park267 *	16	736	14.81	41.08	park119 ***	381	1245.88	15.78	72.7
park119 ***	1473	1400.53	17.31	76.58	park211	14	1255.29	26.32	41.18	park264 ***	321	4180.54	44.84	86.36
park219 ***	1162	1371.9	18.97	69.04	park264 ***	14	5154.29	54.66	80.23	park207	245	3666.07	36.92	99.89
park211	1158	1541.82	21.4	69.71	park119 ***	10	2093.3	21.73	94.84	park275	232	2154.99	22.28	87.46
park230	1013	1293.93	21.89	64.19	park260 ***	10	1385.8	22.9	59.28	park219 ***	228	1760.86	22.87	70.41
park281 **	981	3699.07	41.6	82.98	park259 ***	9	1129.67	46.32	31.32	park259 ***	221	2215.14	30.34	70.78
park276 ***	947	3328.65	36.07	90.3	park276 ***	9	2423.11	34.5	73.52	park117	216	4136.77	79.73	56.63
park133 ***	902	2475.39	31.63	73.69	park307 *	9	1326.44	15.96	88.81	park276 ***	194	3696.61	38.83	88.64
park244 **	886	3320.66	37.79	80.27	park133 ***	8	4262.13	55.92	78.95	park133 ***	167	2809.72	34.36	73.56
park201	822	2344.53	29.19	75.4	park136 *	8	2902.38	32.84	86.2	park281 **	164	2803.21	33.74	71.54
park130	724	1996.19	24.49	72.62	park314 *	8	2944	31.8	86.47	park164	138	758.99	12.16	63.3

*** a park common to three groups; ** a park mostly-walked in Group All & Group 13; * a park only listed in Group 0.

**Table 6 ijerph-19-00395-t006:** List of three types of ‘mostly walked parks’ (from Seoul Urban GIS dataset).

Category	Park ID	Code	Name (Translated from Korean)	Facility Type	Area (m^2^)	Length of Park (m)	avg. Slope (%)	std. Slope	Length of Pedestrian Trail (m)
TYPE MWP-3 a park common to three groups	park260	3-A	Yangjae neighborhood park	Multi-purpose park	32,521	915	0	0	779.3
	park286	3-B	Naegok residential area park	Multi-purpose park	118,397	6295	3.2	4.35	1449.5
	park168	3-C	Eastside Yangjae citizen’s forest	Multi-purpose park	6547	890	2.3	1.25	155.8
	park259	3-D	Seocho Raemian apartment park	Neighborhood park	41,438	1111	8.41	8.82	1431
	park312	3-E	Seocho Nature Hill park	Neighborhood park	98,625	4565	5.67	6.17	204.1
	park122	3-F	Dutbeol park	Multi-purpose park	5876	459	0	0	70.5
	park264	3-G	Yangjae stream park	Multi-purpose park	19,258	1526	1.31	1.36	790.5
	park119	3-H	Banpo neighborhood park	Multi-purpose park	26,092	1110	7.17	6.09	672.5
	park219	3-I	Banpo 1-dong community center	Multi-purpose park	9942	832	0	0	
	park276	3-J	Banpo apartment 1 playground	Multi-purpose park	5483	348	3.08	2.12	137.6
	park133	3-K	Westside Yangjae citizen’s forest	Multi-purpose park	182,632	2091	0.89	0.46	3734
TYPE MWP-2 a park common to two groups, Group All & Group 13	park330	2-A	Seoripool park	Multi-purpose park	652,804	8409	15.35	8.09	5439.9
	park300	2-B	Gomurae children park	Multi-purpose park	18,609	1007	3.43	3.29	581.5
	park204	2-C	Jamwon park	Multi-purpose park	12,963	1162	0.77	0.18	512
	park281	2-D	New Banpo park	Neighborhood park	13,610	1035	0.89	0.54	392.1
	park244	2-E	Dogu-meori park	Multi-purpose park	70,338	1390	15.43	6.93	1333.2
TYPE MWP-1 a park which is only listed in Group 0	park272	1-A	Gaon children park	Children park	3024	396	8.05	5.83	12.1
	park118	1-B	Mido children park	Multi-purpose park	988	127	3.22	1.71	
	park136	1-C	Culture and art park	Multi-purpose park	74,385	1268	1.66	2.14	1429.9
	park307	1-D	Seoul Human Resources Institute	Multi-purpose park	24,121	1407	10.07	4.36	502.7
	park267	1-E	Seocho stadium park	Sports park	18,278	1066	4.4	3.93	60.8
	park248	1-F	Bon-maeul children park	Multi-purpose park	892	124	0	0	
	park314	1-G	Bangbae neighborhood park	Multi-purpose park	265,020	2490	17.62	7.37	362.7

**Table 7 ijerph-19-00395-t007:** Prevalent intensity of parks, finding ‘intensely-walked park’.

Category			Group All					Group 13				
Type-MWP	Park ID	Code	LPA (*n*)	LPA (%)	MVPA (*n*)	MVPA (%)	Prevalent Intensity (avg. = 14.95%)	LPA (*n*)	LPA (%)	MVPA (*n*)	MVPA (%)	Prevalent Intensity (avg. = 19.20%)
MWP-3	park260	3-A	3621	87.60%	514	12.40%	L	866	82.60%	182	17.40%	L
	park286	3-B	1496	89.50%	176	10.60%	L	405	88.60%	52	11.40%	L
	park168	3-C	2431	91.00%	239	9.00%	L	617	86.70%	95	13.30%	L
	park259	3-D	988	86.30%	157	13.70%	L	166	75.10%	55	24.90%	MV
	park312	3-E	1347	90.30%	144	9.70%	L	320	88.90%	40	11.10%	L
	park122	3-F	1427	83.00%	293	17.10%	MV	296	75.30%	97	24.70%	MV
	park264	3-G	996	68.30%	463	31.80%	MV	236	77.40%	69	22.60%	MV
	park119	3-H	1053	85.80%	174	14.20%	L	346	90.80%	35	9.20%	L
	park219	3-I	929	90.40%	99	9.60%	L	175	83.70%	34	16.20%	L
	park276	3-J	566	66.10%	290	33.90%	MV	93	67.40%	45	32.60%	MV
	park133	3-K	498	86.60%	77	13.40%	L	126	87.50%	18	12.50%	L
MWP-2	park330	2-A	3628	87.00%	542	13.00%	L	521	81.40%	119	18.60%	L
	park300	2-B	2874	91.20%	276	8.70%	L	644	95.50%	30	4.50%	L
	park204	2-C	1703	88.50%	221	11.50%	L	302	86.80%	46	13.20%	L
	park281	2-D	361	84.30%	67	15.60%	MV	75	91.50%	7	8.50%	L
	park244	2-E	681	76.90%	205	23.20%	MV	143	57.20%	107	42.80%	MV
MWP-1	park272	1-A	412	80.80%	98	19.20%	MV	92	67.20%	45	32.80%	MV
	park118	1-B	395	94.50%	23	5.50%	L	90	97.80%	2	2.20%	L
	park136	1-C	153	81.40%	35	18.60%	MV	12	92.30%	1	7.70%	L
	park307	1-D	135	65.20%	72	34.80%	MV	50	58.10%	36	41.80%	MV
	park267	1-E	205	91.50%	19	8.40%	L	13	100.00%	0	0.00%	L
	park248	1-F	66	77.60%	19	22.40%	MV	9	56.30%	7	43.80%	MV
	park314	1-G	274	90.40%	29	9.50%	L	88	85.40%	15	14.60%	L

**Table 8 ijerph-19-00395-t008:** Parks grouped by the prevalent intensity of Group All and Group 13.

Prevalent Intensity (Group All/Group 13)	L/L	L/MV	MV/L	MV/MV
Counts	13	1	2	7
park code	3-A	3-D	2-D	3-F
	3-B		1-C	3-G
	3-C			3-J
	3-E			2-E
	3-H			1-A
	3-I			1-D
	3-K			1-F
	2-A			
	2-B			
	2-C			
	1-B			
	1-E			
	1-G			

## Data Availability

Third party data; data were obtained from the mHealth system, WalkOn, and are available with the permission of the Health Promotion Department in the Metropolitan Government of Seoul, and app developer, Swallaby, Inc.

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
