# Peer review of "Differences in Park Walking, Comparing the Physically Inactive and Active Groups: Data from mHealth Monitoring System in Seoul"

_ijerph, 2021, doi:10.3390/ijerph19010395_

Round 1

Reviewer 1 Report

- i don´t think it is a good idea to do a research about elementar and logical things - compare people who are active and make more steps in a day with people who are not active, only walking in park during the day... missing scientific tone.. no connection to urbanism, city planning, I see here a big gap/chance to move the research further - connect it with city planning etc,

Author Response

We would like to explain better our research in this revision.

First of all, research design should be clarified. We describe the research process and illustrate the diagram of research design in Figure 2. 

(Figure 2) Research Design

(Line 130-145) In order to find the differences in walking measures between physically inactive and active people, we designed this research into three processes: 1) selecting sample groups; 2) comparing sample groups; 3) finding certain parks based on the differences between sample groups. First, sample selection classified the physical inactivity level of each user by the number of weeks of regular walking (> 150 minutes/week of moderate-to-vigorous walking activities). Two distinct sample groups, who did/didn’t achieve regular walking every week in research period, were selected.

Second, we compared the walking measures of the sample groups’ park walking and non-park walking. Walking measures of each group, for example, the number of bouts, steps, duration, and cadence, were calculated from the dataset. In the result of this study, the average walking measures of each group were compared, and a significant statistical difference between the two groups was tested by using Mann-Whitney U test.

Third, based on the differences in overall walking activities between each sample groups, it is necessary to figure out the apportioned differences at each park. Considering the health intervention for physically inactive people, we analyze which a certain park has better walking measures.

In order to add the scientific background of this study, a number of literatures on smartphone-based physical activity assessment were explained to make clear the position of our method. We also explain the detailed process of our method, the technical reliability and the possibility of bias.

(Line 43 - 54) “Innovations in physical activity measurement technology [12] and mobile health [13] would be supportive in researches. In particular, walking assessment has been developed from self-reported method such as travel survey, and now objectively-measurements were capable of more quantitative analysis. Global positioning system (GPS), a widely known technology for objectively-measurement of locations and paths, has contributed to analyze the pedestrian behaviors. Combined with geographic information system (GIS), many studies have found the characteristics of walking activity in specific urban area: walking in retail districts [14] and central business district [15]; hiking trails in recreational areas [16]; and park use in urban forest parks [17]. Today, most smartphones are equipped with GPS sensors, and their operating systems detect user’s body movement and accumulate physical activity data. Smartphone-based approach also enables easier physical activity assessment from a larger population than other methods [18, 19].”

(Line 68 - 76) “The reason why this app is suitable for research purposes is that, first, it has been officially used by public health centers in Korea; second, it is free of charge, serviced in Korean, and has more than 500,000 active users (about 1% of national population); and third, it is highly applicable to various smartphone devices because it is based on physical activity assessment algorithm of Android and iOS operating systems. However, its data might be biased because the user percentage is higher in those over 40s and lower under 10s than population. Due to the restriction of personal information, we cannot present the prevalence of ages of our dataset.”

(Line 85-87) “Technical reliability of this data basically follows the confidence level of Android and iOS algorithms that detect physical activity from each device. This is because the raw data of this app has been called from smartphone operating system.”

Including ‘physical inactivity’, the previous manuscript was needed to explain the operational definitions and terms in this study. We put the explanation of terms: physical inactivity, regular walking, park walking, etc.  

(Line 109 - 126)

    • physical inactivity: adults who have less than 150 minutes/week of moderate physical activity or 75 minutes/week of vigorous physical activity, according to the global recommendation of WHO in 2020.
    • regular walking: at least 150 minutes/week of walking activity. This criteria is officially recommended by Ministry of Health and Welfare in Korea to make it easier for the public to understand and promote their physical activities.
    • physically inactive/active sample group: Based on the regular walking of each person, this study categorized people into physically inactive/active sample group.
    • park walking & non-park walking: a walking activity which does/doesn’t pass through park areas in research site. To identify park walking activities, we coded a Point-in-Polygon algorithm in Python to determine whether every GPS points of walking activity were located in park or not.
    • walking measures: quantitative information of walking activity, including accelerometer counts, GPS coordinates, timestamp, etc., were recorded by smartphone device. The application of mHealth system can calculate those numbers into several walking measures, such as steps/bout, duration/bout, and cadence/bout.

We are bit cautious that we over-claim the implication of our results. We don’t think our findings cannot be directly connected to urban planning or park design. For the scope of this journal, ‘environmental research and public health’, we hope this finding as an empirical evidence would be helpful to health intervention for physically inactive people in urban places.

(Line 341 - 347) “Differences in park walking could suggest evidence to distinguish the advantage of parks, with respect to which are easy to make the first park walking, which are favorable to walk longer, and which are effective to increase MVPAs. It could suggest the implication for practice that re-design a park with fewer walking bouts, or run a walk-promoting program at a park with high MVPAs. We hope our collaborative understanding of differences in park walking contributes to providing more supportive built environments for physically inactive people.”

Reviewer 2 Report

The authors present an empirical study worth trying. Some limitations need to be addressed as follows: 

If there is any accuracy index (like Mann-Whitney U test mentioned in section 2), put it in the abstract. Please add “moderate-to-vigorous physical activities” before using its abbreviation (MVPAs).  

Please add a review of previous research on “Innovations in physical activity measurement”. Also pls explain the novelty of this work. 

In section Research Data and Methods, the source of park data should be provided.  

The classifying standard between park-walking group and non-park-walking group need to be illustrated, is it classified by path, or user? It is unclear whether park-walking group refers to the collection of park-walking paths or the group of park-walking people. How many park-walking are required to be classified into park-walking group? 

The overall analysis should be deepened. Apart from such measurements as the number of bouts, steps, duration, and cadence, deeper understandings of the walking pattern is expected. The division of inactive and active groups could be improved, for instance those between 0-13 may also be analyzed. Clustering method could be applied to distinguish the different type of path or user or walking time, and regression models might help in determining the factors influencing the walking, such as the park location/size, walking time, speed etc. 

More reference should be included.  The following are some examples:

   Using participatory GIS to measure physical activity and urban park benefits

   Active-transport walking behavior: destinations, durations, distances

  Do built environments affect pedestrians' choices of walking routes in retail districts? A study with GPS experiments in Hongdae retail district in Seoul, South Korea.

   Evaluating the structure and use of hiking trails in recreational areas using a mixed GPS tracking and graph theory approach

   Spatial distributions and use patterns of user groups in urban forest parks: An examination utilizing GPS tracker

   Personal mobility pattern mining and anomaly detection in the GPS era

   Analysing hiker movement patterns using GPS data: Implications for park management

   Travel mode detection based on GPS track data and Bayesian networks

   A Hotspot of Walking in and around the Central Business District: Leveraging Coarsely Geocoded Routinely Collected Data

Author Response

Thanks to your review, we were able to supplement the limitations of this study.

First, we add some correction to abstract: p-values for statistical significance, and a full form of abbreviation.

(Line 19) “...had more steps (p = .021), longer time (p = .008), and higher intensity (p < .001) of walking...”

(Line 22) “moderate-to-vigorous physical activities(MVPAs)”

Examples of reference you suggested to us were so helpful that we can update our theoretical background to innovations in physical activity measurement.

(Line 44 - 54) “In particular, walking assessment has been developed from self-reported method such as travel survey, and now objectively-measurements were capable of more quantitative analysis. Global positioning system (GPS), a widely known technology for objectively-measurement of locations and paths, has contributed to analyze the pedestrian behaviors. Combined with geographic information system (GIS), many studies have found the characteristics of walking activity in specific urban area: walking in retail districts (Hahm et al., 2017) and central business district (Mazumdar et al., 2020); hiking trails in recreational areas (Taczanowska et al., 2014); and park use in urban forest parks (Zhai et al., 2018). Today, most smartphones are equipped with GPS sensors, and their operating systems detect user’s body movement and accumulate physical activity data.”

Among many studies for physical activity assessment in specific urban areas, the position of our research method could be further clarified in Chapter 2, Research Data and Methods. As a smartphone-based physical activity measurement, we explain more about the uniqueness of our method: the reason why we chose this application for study; the process how the user’s physical activities were transmitted through research dataset; and the reliability that smartphone-based data had.

(Line 69-73) “The reason why this app is suitable for research purposes is that, first, it has been officially used by public health centers in Korea; second, it has more than 500,000 active users; and third, it is highly applicable to other smartphones because it is based on physical activity assessment algorithm of Android and iOS operating systems.”

(Line 82-84) “Technical reliability of this data basically follows the confidence level of Android and iOS algorithms that detect physical activity from each device. This is because the raw data of this app has been called from smartphone operating system.”

(Line 90-95) “De-identification process must be applied on research data so that individuals cannot be specified, according to the Personal Information Protection Act in Korea. In order to determine the user’s physical inactivity, the number of week that the user had achieved regular walking in thirteen weeks of research period. Park facilities in GIS dataset, shown in Figure 1 were used to distinguish park walking...”

Including the source of park dataset, we explain three main datasets of this study. And the definitions of terms, such as ‘park walking’, are illustrated in order to reduce misunderstanding. We admit that the expression ‘park walking group or non-park walking group’ could be confused, and we don’t want to use them. The definition of ‘park walking’ only classify the walking activity that passed through a park. ‘Group’ is meant that people who have been categorized by the number of regular walking weeks. Some people in a certain sample group might have both park walking activities and non-park walking activities, and the other in same sample group only have non-park walking activities. However, this research does not compare those people in same group, rather compare people in different groups. In order to avoid confusion from mixing the definition of walking activities and groups, thus, we edit Table 2, separating the column into new columns, ‘sample’ and ‘activity’.

(Line 85-88) “Research data of this study are mainly composed of three types of information: smartphone-based physical activity data for walking activities; smartphone-based regular walking data for physical inactivity level of subjects; and Seoul urban GIS data for park facilities.”

(Table 2) add new columns, “sample” and “activity”

We agree that, as you suggested, there is a need to proceed with a deeper analysis of walking patterns. Instead of selecting two sample groups, we tried to set several index of walking measures to evaluate all subjects and cluster them between similar patterns. Although our efforts were made to statistically examine the differences in park walking between such clusters, we haven’t found any significant results yet. We are working with a new research design for that approaches. The current findings of this study give us evidences about the differences in elemental walking measures, such as bouts, steps, duration, and cadence, between these two sample groups which were expected to have distinctly different level of physical inactivity. We suggest a new figure of study design to help understanding our study.

(Figure 2) add new diagram for research design

(Line 124 - 130) “In order to find the differences in walking measures between physically inactive and active people, we designed this research into three processes: 1) selecting sample groups; 2) comparing sample groups; 3) finding certain parks based on the differences between sample groups. First, sample selection classified the physical inactivity level of each user by the number of weeks of regular walking (> 150 minutes/week of moderate-to-vigorous walking activities). Two distinct sample groups, who did/didn’t achieve regular walking every week in research period, were selected.”

Reviewer 3 Report

The manuscript is relevant, very actual and interesting. The authors basically combine the viewpoints of medical science, GIS and, in part, urban planning in their interdisciplinary approach.

However, for me, the study focuses too much on the circumstances and context of a selected district of Seoul, so it would be worth mentioning more about certain additional factors so therefore non-Korean, foreign readers could better interpret it.
For example, various additional factors may affect the prevalence of the analysed application (e.g. mobile penetration according to different age groups relevant in the walking behaviour aspects from the study, which may not only be indirectly cited but you need also to detail it in the body text, I suppose, and, in addition, e.g. whether the application is free of charge, or which mobile operating systems are needed for using that or in what languages it operates).

These can also be described because the manuscript does not give me completely clear that what kind of anonymized data you have obtained from the mobile users of the application overall (I am a European and here we have very strict rules in this topic as GDPR sometimes makes researches quite difficult here).

Although you mention the limitations of your research in the manuscript, which is correct, I suppose, I  have some questions about the limitations of your study. The answer for them may help the reader to interpret the manuscript even more scientifically.

How accurate was the geolocalisation of anonymous mobile users in space and time during their stay in the parks, and if was there any factor that could have influenced measurement accuracy in space and time? (e.g. weaker reception of mobile signs at certain spatial points of parks, or perhaps too many simultaneous users in the mobile operator's cell at the same time)?

What population was the purpose of the sampling of study (included the people living in the district or anyone who was in the parks inspected at the time of the study)?

Why did you choose the given district in Seoul?

And, finally, according to you, how much can your results could generalised for the cases of other districts of Seoul, or for other urban districts outside Seoul in the Republic of Korea?

Author Response

We are grateful that your review allows us to develop our limitations of manuscript.

The research site and method are not quite globally popular and it can be relevant with local contexts and circumstances, it is necessary to fully explain the possibility of bias. As you mentioned, we add the detail and limitation of our application and methods.

(Line 68 - 76) “The reason why this app is suitable for research purposes is that, first, it has been officially used by public health centers in Korea; second, it is free of charge, serviced in Korean, and has more than 500,000 active users (about 1% of national population); and third, it is highly applicable to various smartphone devices because it is based on physical activity assessment algorithm of Android and iOS operating systems. However, its data might be biased because the user percentage is higher in those over 40s and lower under 10s than population. Due to the restriction of personal information, we cannot present the prevalence of ages of our dataset.”

(Line 80) “The research site, Seocho-gu district, .... , and the largest number of pedestrians.”

(Line 85 - 87) “Technical reliability of this data basically follows the confidence level of Android and iOS algorithms that detect physical activity from each device. This is because the raw data of this app has been called from smartphone operating system.”

We agree with the insufficient explanation of the data privacy issue, despite the data of this study were mainly assessed by smartphone. Following the Personal Information Protection Act in Korea which was revised in 2020 under the influence of European GDPR, existing datasets from Seoul mHealth monitoring system were provided for our research after de-identification process..

(Line 91 - 97) “Our dataset of walking activities, which was existing dataset from Seoul mHealth monitoring system, has information of pseudonymized userID, start time, end time, the number of steps, and GPS coordinates of its path (Appendix, Table A1). De-identification process must be applied on research data so that individuals cannot be specified, according to the Personal Information Protection Act in Korea. In order to determine the user’s physical inactivity, the number of week that the user had achieved regular walking in thirteen weeks of research period.”

It is difficult to estimate the accuracy of our GPS data from different smartphone devices at different location, but we would like to cite relevant sources:

“Overall average horizontal position accuracy of the iPhone 6 (7-13m) is consistent with the general accuracy level observed of recreation-grade GPS receivers in potential high multi-path environments.” (Merry, K., & Bettinger, P. (2019). Smartphone GPS accuracy study in an urban environment. PloS one, 14(7), e0219890.)

“GPS-enabled smartphones are typically accurate to within a 4.9m radius under open sky (source from ION.org). However, their accuracy worsens near buildings, bridges, and trees.” (GPS.gov, Official US government information about the GPS and related topics, https://www.gps.gov/systems/gps/performance/accuracy/)

Our dataset from mHealth system database was filtered by the condition of our research range. Thus, the population of data is consisted of every users who had at least one walking activity in research period and research site.

(Line 83 - 85) “The dataset of this study was composed of 22744 app users who recorded 550234 bouts of walking activity in Seocho-gu district during 13 weeks, from September to November 2019.“

Generalizing our results for the cases in other area would be difficult to say. We need further researches on other districts or other cities. This study contributes to suggest how to enlarge the sample size by using smartphone-based assessment, how to identify walking activities on certain urban places, and how to compare their measures between different subject groups. It is an initial step to explore the association between walking on urban facilities and the level of physical inactivity. We are working with several regression models to find spatial and environmental features that might affect physically inactive people. We hope to continue and broaden our research.

Reviewer 4 Report

The manuscript " Differences in Park Walking, Comparing the Physically Inactive and Active Groups: Data from mHealth Monitoring System in Seoul" deals with the topic of walkability in urban parks. The topic of walkability and healthy open spaces (parks) is meant for deep research as one of the hot topics in urban studies. The case of the Seocho-gu district in Seoul could be one typical urban regeneration case worth for study. However, the manuscript has some serious limitations and major flaws that need further revision as follow:

  1. Definitions of basic concepts like physical inactivity need to be clarified, for a clear understanding of the study meaning of this article.
  2. I understand innovation in method, which use mobile database and makes no use of demographic factors and park characteristics. Do we really need/have to threaten 130 urban spaces (parks) as they have some characteristics of walkability? Are all needs of physical inactivity people same or similar?
  3. Coherence in research design: authors highlight, in the introduction, that is little is known about physically inactive people who highly need the health intervention, but applications of this research to the issue were not discussed.
  4. Research design: there is no discussion.
  5. What is a scientific contribution? Is it a research method or a substantial contribution to the concepts? I failed to understand this.
  6. Method: some parts of the results explained the research method. For example lines, 94-104 are not results by the authors. The statements in these lines make material for method and introduction.
  7. Method: the context of the research: The method section lacks an explanation of the reasons why the authors choose the WalkOn App. We found only in conclusion that Korea has 95% of smartphone ownership. But, what about Seoul and what about the district in the examination? Do all the people on the WalkOn App use it?
  8. Method: For me, it was not clear, are 13 weeks adequate time? At which moment authors stopped the measurement of other samples - people who does not belong to the two distinctive sample groups, which had maximum and minimum of a regular walking week.
  9. What is the authors’ contribution? The authors did not gather data. They did not discuss how the categorisation could be used to contribute to physically inactive people.
  10. Valuable aspects: clear goals and classification themselves are useful if the authors contextualise them well in a scientific framework. Moreover, it is a practical example of applying a smartphone-based monitoring system to health policy with collaborative understanding. These aspects of different interests should be highlighted.

I sincerely encourage the authors to improve and resubmit the manuscript.

Author Response

With your detailed comments, the authors and I were able to revise the limitation of study and improve the manuscript. Major revision points, as you mentioned, would be explained as follows:

  1. Including ‘physical inactivity’, the previous manuscript was needed to explain the operational definitions and terms in this study. We put the explanation of terms: physical inactivity, regular walking, park walking, etc.

(Line 109 - 126) physical inactivity: adults who have less than 150 minutes/week of moderate physical activity or 75 minutes/week of vigorous physical activity, according to the global recommendation of WHO in 2020.

regular walking: at least 150 minutes/week of walking activity. This criteria is officially recommended by Ministry of Health and Welfare in Korea to make it easier for the public to understand and promote their physical activities.

physically inactive/active sample group: Based on the regular walking of each person, this study categorized people into physically inactive/active sample group.

park walking & non-park walking: a walking activity which does/doesn’t pass through park areas in research site. To identify park walking activities, we coded a Point-in-Polygon algorithm in Python to determine whether every GPS points of walking activity were located in park or not.

walking measures: quantitative information of walking activity, including accelerometer counts, GPS coordinates, timestamp, etc., were recorded by smartphone device. The application of mHealth system can calculate those numbers into several walking measures, such as steps/bout, duration/bout, and cadence/bout.

  1. Despite the different contexts and characteristics of 130 park facilities, we consider them as environmental resources that our city has. Although some facilities have less or no contribution for public health (actually, there were several parks which had no walking activities), it is possible to estimate walking measures from observed data, even though they were too small. We don’t insist that every park facilities should promote physical activity, but comparing quantitative measures of each park would be helpful to figure out empirical impacts on people’s walking.

(Line 142 - 145) Third, based on the differences in overall walking activities between each sample groups, it is necessary to figure out the apportioned differences at each park. Considering the health intervention for physically inactive people, we analyze which a certain park has better walking measures.

  1. We are bit cautious that we over-claim the implication of our results, but we hope this finding as an empirical evidence would be helpful to health intervention for physically inactive people. We expect several examples of health intervention, for example, park program or park planning.

(Line 341 - 347) “Differences in park walking could suggest evidence to distinguish the advantage of parks, with respect to which are easy to make the first park walking, which are favorable to walk longer, and which are effective to increase MVPAs. It could suggest the implication for practice that re-design a park with fewer walking bouts, or run a walk-promoting program at a park with high MVPAs. We hope our collaborative understanding of differences in park walking contributes to providing more supportive built environments for physically inactive people.”

  1. To help understand the research design, we newly illustrate the figure 2, diagram of research design, and explain the research processes.

(Line 130-145) In order to find the differences in walking measures between physically inactive and active people, we designed this research into three processes: 1) selecting sample groups; 2) comparing sample groups; 3) finding certain parks based on the differences between sample groups. First, sample selection classified the physical inactivity level of each user by the number of weeks of regular walking (> 150 minutes/week of moderate-to-vigorous walking activities). Two distinct sample groups, who did/didn’t achieve regular walking every week in research period, were selected.

Second, we compared the walking measures of the sample groups’ park walking and non-park walking. Walking measures of each group, for example, the number of bouts, steps, duration, and cadence, were calculated from the dataset. In the result of this study, the average walking measures of each group were compared, and a significant statistical difference between the two groups was tested by using Mann-Whitney U test.

Third, based on the differences in overall walking activities between each sample groups, it is necessary to figure out the apportioned differences at each park. Considering the health intervention for physically inactive people, we analyze which a certain park has better walking measures.

(Figure 2) Research Design

  1. This study contributes to suggest how to enlarge the sample size by using smartphone-based assessment, how to identify walking activities on certain urban places, and how to compare their measures between different subject groups. It is an initial step to explore the association between walking on urban facilities and the level of physical inactivity. We are working with several regression models to find spatial and environmental features that might affect physically inactive people. Generalizing our results for the cases in other area would be difficult to say. We need further researches on other districts or other cities. We hope to continue and broaden our research.
  2. We deleted that sentences in the results part which were not our novel findings.
  3. We write the reason why we chose the research site, period, and method. We also add the possibility of bias and technical reliability of data source, with support of further literature reviews on smartphone-based physical activity assessment.

(Line 68 - 76) “The reason why this app is suitable for research purposes is that, first, it has been officially used by public health centers in Korea; second, it is free of charge, serviced in Korean, and has more than 500,000 active users (about 1% of national population); and third, it is highly applicable to various smartphone devices because it is based on physical activity assessment algorithm of Android and iOS operating systems. However, its data might be biased because the user percentage is higher in those over 40s and lower under 10s than population. Due to the restriction of personal information, we cannot present the prevalence of ages of our dataset.”

(Line 85-87) “Technical reliability of this data basically follows the confidence level of Android and iOS algorithms that detect physical activity from each device. This is because the raw data of this app has been called from smartphone operating system.”

(Line 44 - 53) “In particular, walking assessment has been developed from self-reported method such as travel survey, and now objectively-measurements were capable of more quantitative analysis. Global positioning system (GPS), a widely known technology for objectively-measurement of locations and paths, has contributed to analyze the pedestrian behaviors. Combined with geographic information system (GIS), many studies have found the characteristics of walking activity in specific urban area: walking in retail districts [14] and central business district [15]; hiking trails in recreational areas [16]; and park use in urban forest parks [17]. Today, most smartphones are equipped with GPS sensors, and their operating systems detect user’s body movement and accumulate physical activity data.”

  1. We thought the period of dataset were appropriate because people would have a lot of walking activities in that period. And because the data of this research were provided from the existing database of mHealth system, the filtering range of data could not be easily extended or modified. If some users joined before/after the research period, they cannot be added to the research subjects.

(Line 76 - 78) The period of the dataset, from September to November 2019, was the autumn season with abundant outdoor activities and the season before the outbreak of COVID-19, that might have brought an impact on people’s outdoor activities.

  1. We didn’t gather the raw data, but analyze it. We set the process and write scripts to categorize the physically inactive/active sample group based on regular walking dataset. We identified which walking activities had passed through the geospatial boundaries of parks. Also, we calculated walking measures from raw activity dataset. We statistically tested the homogeneity of walking measures between two sample groups. We estimated average walking measures at each park, and compare the sample groups. Authors’ contribution is to set the whole process to analyze the raw data.
  2. This study contributes to suggest how to enlarge the sample size by using smartphone-based assessment, how to identify walking activities on certain urban places, and how to compare their measures between different subject groups. It is an initial step to explore the association between walking on urban facilities and the level of physical inactivity. We are working with several regression models to find spatial and environmental features that might affect physically inactive people. Generalizing our results for the cases in other area would be difficult to say. We need further researches on other districts or other cities. We hope to continue and broaden our research.

Round 2

Reviewer 1 Report

thank yu for improving the paper, i kniw it is hard sometimes to satisfy all the reviewers.

Author Response

Your valuable comments motivated us to develop this manuscript.

It was sometimes difficult, as you mentioned, but we believe that our responsibility is to be faithful to the entire review process.

We appreciate you for reviewing our manuscript. 

Reviewer 2 Report

You've done an insteresting and encouraging research and I have no more suggetions.

Author Response

Without your careful review and significant comments, we could not have improved our manuscript more scientifically.

We look forward to this study contributing to the discussion on physical activity and urban environments. Thank you very much.